# Regional differences in short stature in England between 2006 and 2019: A cross-sectional analysis from the National Child Measurement Programme

**Joanna Orr**[1]*, **Joseph Freer**[1], **Joan K. Morris**[2], **Caroline Hancock**[3], **Robert Walton**[1], **Leo Dunkel**[1], **Helen L. Storr**[1], **Andrew J. Prendergast**[1]

**1** Queen Mary University of London, London, United Kingdom, **2** St George's University of London, London, United Kingdom, **3** Public Health England, London, United Kingdom

* joanna.orr@qmul.ac.uk

**Data Availability Statement:** Limited versions of the datasets used in these analyses are available from the UK Data Service (2006 to 2012, https://

## Abstract

### Background

Short stature, defined as height for age more than 2 standard deviations (SDs) below the population median, is an important indicator of child health. Short stature (often termed stunting) has been widely researched in low- and middle-income countries (LMICs), but less is known about the extent and burden in high-income settings. We aimed to map the prevalence of short stature in children aged 4–5 years in England between 2006 and 2019.

### Methods and findings

We used data from the National Child Measurement Programme (NCMP) for the school years 2006–2007 to 2018–2019. All children attending state-maintained primary schools in England are invited to participate in the NCMP, and heights from a total of 7,062,071 children aged 4–5 years were analysed. We assessed short stature, defined as a height-for-age standard deviation score (SDS) below −2 using the United Kingdom WHO references, by sex, index of multiple deprivation (IMD), ethnicity, and region. Geographic clustering of short stature was analysed using spatial analysis in SaTScan. The prevalence of short stature in England was 1.93% (95% confidence interval (CI) 1.92–1.94). Ethnicity adjusted spatial analyses showed geographic heterogeneity of short stature, with high prevalence clusters more likely in the North and Midlands, leading to 4-fold variation between local authorities (LAs) with highest and lowest prevalence of short stature. Short stature was linearly associated with IMD, with almost 2-fold higher prevalence in the most compared with least deprived decile (2.56% (2.53–2.59) vs. 1.38% (1.35–1.41)). There was ethnic heterogeneity: Short stature prevalence was lowest in Black children (0.64% (0.61–0.67)) and highest in Indian children (2.52% (2.45–2.60)) and children in other ethnic categories (2.57% (2.51–2.64)). Girls were more likely to have short stature than boys (2.09% (2.07–2.10) vs. 1.77% (1.76–1.78), respectively). Short stature prevalence declined over time, from 2.03% (2.01–2.05) in 2006–2010 to 1.82% (1.80–1.84) in 2016–2019. Short stature

beta.ukdataservice.ac.uk/datacatalogue/studies/study?id=7567) and NHS Digital (2013 to 2019, https://digital.nhs.uk/data-and-information/publications/statistical/national-child-measurement-programme). Full datasets cannot be shared publicly to protect the anonymity of children participating in the NCMP.

**Funding:** This study was funded by Barts Charity (grant MRC0219). AJP is funded by Wellcome (grant 108065/Z/15/Z). The funders had no role in the study design, data collection and analysis, decision to publish, or preparation of the manuscript.

**Competing interests:** The authors have declared that no competing interests exist.

**Abbreviations:** CI, confidence interval; IMD, index of multiple deprivation; IRAS, Integrated Research Application System; LA, local authority; LMIC, low- and middle-income country; NCMP, National Child Measurement Programme; PHE, Public Health England; RR, relative risk; SD, standard deviation; SDS, standard deviation score; STROBE, Strengthening the Reporting of Observational Studies in Epidemiology.

declined at all levels of area deprivation, with faster declines in more deprived areas, but disparities by IMD quintile were persistent. This study was conducted cross-sectionally at an area level, and, therefore, we cannot make any inferences about the individual causes of short stature.

## Conclusions

In this study, we observed a clear social gradient and striking regional variation in short stature across England, including a North–South divide. These findings provide impetus for further investigation into potential socioeconomic influences on height and the factors underlying regional variation.

## Author summary

### Why was this study done?

- Short height for age can be a sign that there are underlying health conditions or adverse socioeconomic circumstances in young children.

- Research into children who have short stature in low- and middle-income countries (LMICs) has found numerous associations with poorer lifelong health and education outcomes.

- The prevalence and characteristics of children with short stature in England have not previously been investigated.

### What did the researchers do and find?

- We used data from 7,062,071 children aged 4 and 5 who were measured as part of the National Child Measurement Programme (NCMP) to calculate percentages of children with short stature by region, ethnicity, sex, and area deprivation.

- We used regional analyses to assess whether there were hotspots of short stature around the country.

- We found that 1.93% of children had short stature, and this was strongly related to area deprivation, with more deprived areas having higher rates of short stature.

- Short stature was also less likely in children who were Black African or Caribbean and more likely in Indian children or children whose ethnicity was coded as "Other."

- There were various short stature hotspots in England, and most of these were concentrated in the North and Midlands of the country.

### What do these findings mean?

- Short stature is relatively uncommon in England.

- However, short stature is strongly related to area-level deprivation, with children in the most deprived areas nearly twice as likely to be short as children in the least deprived areas.

- Further investigation into why children in poorer areas are shorter is important.

## Introduction

Linear growth is an important indicator of a child's health. Poor early life growth is associated with impaired physical, neurodevelopmental, and educational outcomes, which hamper children's abilities to survive and thrive [1], and increase later life risk of chronic disease and premature mortality [2]. Stunting, which is defined as a height for age more than 2 standard deviations (SDs) below the reference population median, is a term generally confined to low- and middle-income countries (LMICs). Short stature may therefore be a hidden problem in high-income countries, particularly in economically deprived areas, and an overlooked marker of child well-being.

Because the focus of UK public health programmes on growth is on body mass index, there are no recent data describing the prevalence of child short stature. Surveys in 2011 and 2017, and a roundtable discussion in 2014 conducted by The Patients Association, highlighted anecdotal evidence of an increasing burden of child undernutrition and identified this as a public health priority [3,4]. Moreover, the "dual burden" of stunting and overweight is increasingly being recognised globally [5].

In England, children are measured at ages 4 to 5 and 10 to 11 years through the National Child Measurement Programme (NCMP), which was introduced in 2006 to assess overweight and obesity in primary school children [6]. Data on obesity prevalence and trends are presented in user-friendly dashboards, which include aggregated data by region, ethnicity, and index of inequality [7]. Although the heights of all participating children are collected, there are no equivalent dashboards for prevalence or trends in short stature, and children with linear growth faltering are not routinely identified or directed to health services through the NCMP.

We set out to leverage these national data on height in early childhood using the 13 years of available data to map short stature prevalence across England. Our hypothesis was that there are geographical hotspots of short stature in England, which are concentrated in socioeconomically deprived areas.

## Methods

This study is reported as per the Strengthening the Reporting of Observational Studies in Epidemiology (STROBE) guidelines (S1 Text). The study forms part of a wider project on child growth and development. The original analysis plan, first presented as part of the funding application for this project, is presented in S2 Text.

### Study sample

We used available data from children aged 4 to 5 years in the NCMP for 13 school years between 2006 and 2019. All children attending a state-maintained primary school in the country (94% in 2010 [8]) are invited to participate in the NCMP. The programme has a very high (93%) average participation rate [6].

## NCMP data collection

Detailed protocol descriptions are available for all NCMP procedures [9]. Briefly, children's height and weight were measured by a school nurse or other trained staff member in each participating school using calibrated weight scales and stadiometers. Height and weight were measured in centimetres and kilogrammes to the first decimal place, respectively. Children who declined to participate, or whose parents withdrew them from the programme, were not measured. Children with known growth disorders, Down syndrome, and children who could not stand unaided or could not be measured accurately were measured, but their results were not uploaded to the NCMP system. NCMP data were validated at source, where records with missing mandatory fields were rejected, and records with improbable fields were flagged for the data provider to check before submission. Details on error and warning ranges for height and weight have been published elsewhere [10].

Data were available for 13 years between school years 2006 to 2007 and 2018 to 2019. NCMP data collection is conducted by local authorities (LAs), overseen by Public Health England (PHE), and data are managed by NHS Digital. NHS Digital makes a limited version of the NCMP dataset publicly available for analysis, which does not contain data that could lead to the identification of any child [11]. Access to the full NCMP dataset for the current analysis was obtained through collaboration with PHE, and analyses were conducted at PHE to comply with best data protection practices. Integrated Research Application System (IRAS) approval was not required for the current study, which was registered with the Clinical Governance Department at Barts Health NHS Trust.

## Measures

For the current analysis, height-for-age standard deviation scores (SDSs) were derived for each child using UK WHO reference values [12] using the zanthro package in Stata 15 [13]. Children with SDS above 5 or below −5 were excluded. Short stature was defined a height SDS below −2.0. This cutoff was chosen for comparability with international definitions of stunting. Model covariates included ethnicity and index of multiple deprivation (IMD). Ethnicity was collected by schools for each child and was recoded for this analysis into 6 categories based on the UK government census categorisation: White (White British and White Other); Black (Black African, Black Caribbean, and Black Other); Indian; Pakistani or Bangladeshi; Mixed; and Other ethnicity. The IMD is a composite, area-level measure of relative deprivation. It comprises 7 measures of deprivation including employment deprivation, income deprivation, and crime. The IMD ranks all small areas (lower super output areas) in the country from most deprived (#1) to least deprived (#34,753 in 2019). NCMP provides an IMD decile for each child based on their postcode; we use both IMD decile and quintile in our analysis.

## Statistical analyses

Prevalence of short stature by sex, region, ethnicity, IMD, and time period was examined using Stata 15, and between-group differences were evaluated using chi-squared tests. We assessed whether change over time in short stature prevalence was consistent across levels of area deprivation by examining differences in rates of change by IMD quintile. We fitted a logistic regression model of the probability of short stature by IMD and NCMP year and included an IMD#year interaction term. We visualised change over time using Equiplot, a method for comparing inequality between factors such as time periods [14]. Spatial analyses were conducted using SaTScan [15] to assess geographic clustering of short stature. SaTScan analyses geographical data to identify disease or other event clusters and tests for statistical significance. The software scans a geographical area using a circular window, which is then

positioned at each point in a given geographical grid and varies in size from 0 to an upper limit predetermined by the researcher. A null hypothesis of equal risk inside and outside the circular window is tested using likelihood ratios. The most probable cluster is chosen by this scanning system as that with the highest likelihood ratio. The software then systematically scans for the next most probable cluster. Geographic aggregation units for all spatial analyses were lower tier LAs. In England, lower tier LAs include LA districts, unitary authorities, metropolitan districts, and London boroughs. We scanned the data unadjusted, then adjusted for ethnicity and adjusted for both ethnicity and IMD. A Bernoulli probability model for count data was used; model specifications and details of the adjustment strategy are given in Tables A–D in S3 Text. We initially analysed data for all years and then further assessed whether the clusters identified were consistent over time by analysing 4 NCMP time periods separately (school years 2006 to 2010; 2010 to 2013; 2013 to 2016; and 2016 to 2019). We present all statistically significant clusters (threshold $P < 0.05$). $P$ values were determined by log likelihoods being greater than a critical value established for each model through standard Monte Carlo and Gumbel approximation [15].

### Supplemental analyses

We repeated all analyses using very short stature, defined as height $< -2.67$ SDS (equivalent to $< 0.4$th percentile), which is used as a referral cutoff for investigation of short stature in the UK [16]. Due to the exclusion of some of the sample from spatial analyses because of missing ethnicity and IMD data, a sensitivity analysis was conducted running an unadjusted SaTScan model on the full dataset to check for bias. We also examined the percentages of children by sex, ethnicity, and region in our dataset against census and projections data from 2011 to -2012 to assess whether our sample matched the population.

### Ethical approval

The current study is a secondary analysis of NCMP data, and, therefore, ethical approval was not required. The study was registered with the Clinical Governance Department at Barts Health NHS Trust.

## Results

### Prevalence of short stature

A total of 7,299,208 reception-aged children participated in the NCMP between 2006 and 2019. NCMP response rates increased from 83% in the 2006 to 2007 school year to 95% in 2018 to 2019. From this population, we included 7,062,071 children aged 4 to 5 years at measurement who had valid locality data in our analysis of country prevalence. Of these, 5,765,707 (82%) had valid ethnicity and IMD data and were included in unadjusted and adjusted spatial models of short stature clustering. Details of inclusion in the sample are given in Fig 1. Missingness by sample characteristics is given in S1 Table. Ethnicity was missing in 18% of children, while IMD was missing in very few children (0.01%). Missingness was explained in part by improvements in NCMP data validation processes over time, with a larger percentage of missing data in the first time period. Missingness was also related to region but did not appear to be related to height or short stature. We also examined 2011 Census data as well as data projections for child population numbers by region and found that the percentages of each sex, ethnic group, and region matched the population very closely (S2 Table).

Table 1 shows sample characteristics and prevalence of short stature. The mean (SD) age of children in the sample was 60.0 (4.0) months, and the mean (SD) height was 109.6 (5.1) cm. In

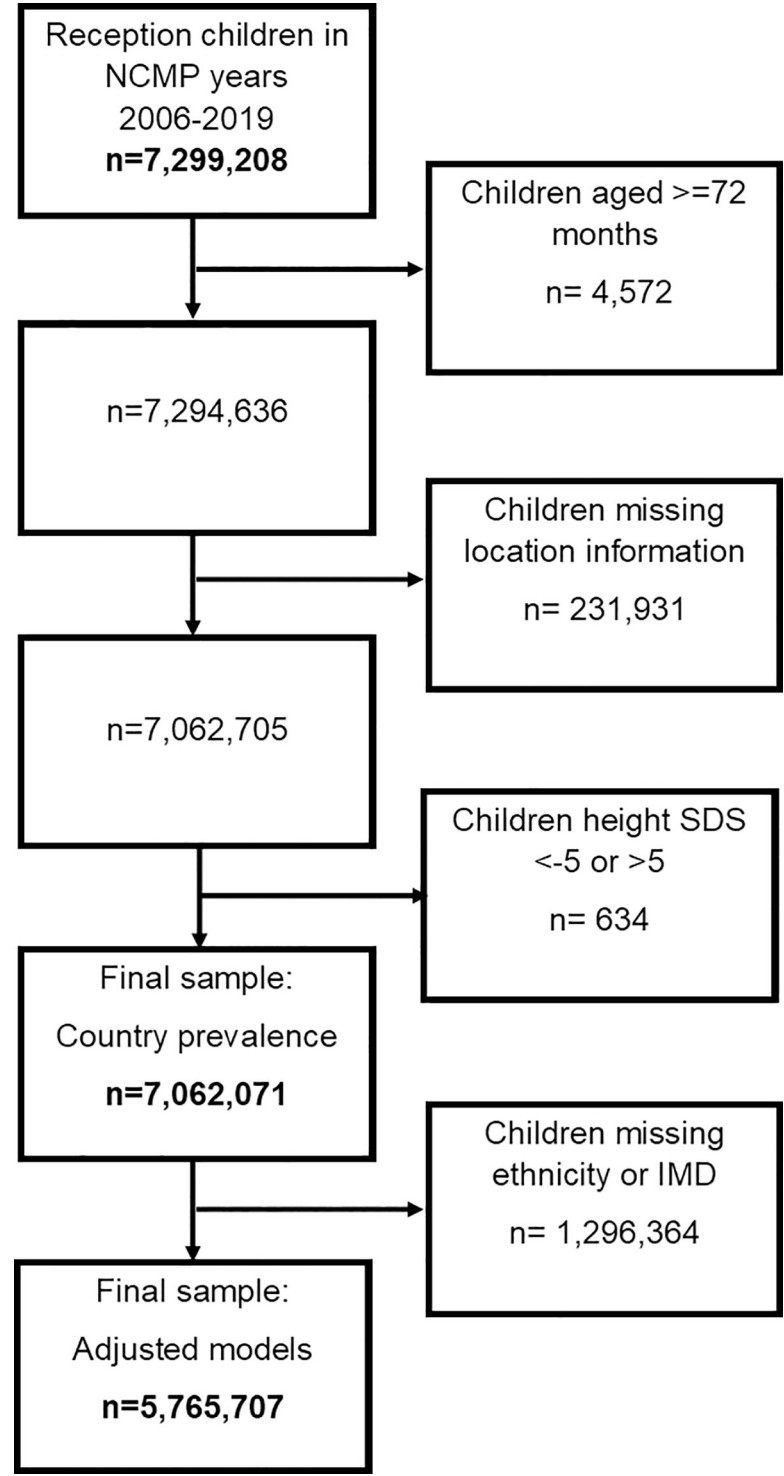

**Fig 1. Sample inclusion and exclusion flowchart.** IMD, index of multiple deprivation; NCMP, National Child Measurement Programme; SDS, standard deviation score.

the period between 2006 and 2019, 1.93% (95% confidence interval (CI): 1.92 to 1.94) of children aged 4 to 5 years were short for their age in England. Girls had a higher prevalence of short stature, with 2.09% (2.07 to 2.10) of girls compared with 1.77% (1.76 to 1.78) of boys

**Table 1. Short stature (height <−2.0 SDS) prevalence by sample characteristics (*n* = 7,062,071).**

| Characteristic | Mean (SD) % (*n*) | Short stature, % [95% CI] |
|---|---|---|
| Age in months | | |
| Mean (SD) | 60.0 (4.0) [60.0; 60.0] | - |
| Height | | |
| Mean height cm (SD) [95% CI] | 109.6 (5.1) [109.6; 109.6] | - |
| Mean height SDS (SD) [95% CI] | 0.11 (1.0) [0.11; 0.11] | - |
| Sex % (*n*) | | |
| Male | 51 (3,608,608) | 1.77 [1.76; 1.78] |
| Female | 49 (3,453,463) | 2.09 [2.07; 2.10] |
| Government Office Region % | | |
| North East | 5 (351,119) | 2.12 [2.08; 2.17] |
| North West | 14 (974,826) | 2.14 [2.12; 2.17] |
| Yorkshire and the Humber | 10 (719,264) | 2.18 [2.15; 2.22] |
| East Midlands | 9 (597,755) | 2.08 [2.04; 2.11] |
| West Midlands | 11 (788,566) | 2.05 [2.02; 2.08] |
| East of England | 11 (795,006) | 1.89 [1.86; 1.92] |
| London | 16 (1,110,081) | 1.57 [1.55; 1.60] |
| South East | 15 (1,080,309) | 1.74 [1.72; 1.77] |
| South West | 9 (645,145) | 1.86 [1.83; 1.89] |
| Ethnicity % (*n*) | | |
| White British and White Other | 62 (4,380,202) | 1.95 [1.94; 1.96] |
| Black African, Caribbean, and Other | 5 (314,293) | 0.64 [0.61; 0.67] |
| Indian | 2 (168,896) | 2.52 [2.45; 2.60] |
| Pakistani and Bangladeshi | 5 (349,060) | 2.18 [2.13; 2.23] |
| Mixed | 4 (304,769) | 1.58 [1.53; 1.62] |
| Other | 4 (248,874) | 2.57 [2.51; 2.64] |
| Missing | 18 (1,295,977) | 1.96 [1.94; 1.98] |
| IMD (decile)[a] % (*n*) | | |
| 1 | 14 (984,279) | 2.56 [2.53; 2.59] |
| 2 | 12 (860,767) | 2.24 [2.21; 2.27] |
| 3 | 11 (768,760) | 2.09 [2.06; 2.12] |
| 4 | 10 (698,317) | 2.00 [1.96; 2.03] |
| 5 | 9 (655,926) | 1.88 [1.84; 1.91] |
| 6 | 9 (625,186) | 1.75 [1.72; 1.79] |
| 7 | 8 (599,197) | 1.70 [1.67; 1.74] |
| 8 | 9 (609,677) | 1.62 [1.59; 1.65] |
| 9 | 9 (623,088) | 1.51 [1.48; 1.54] |
| 10 | 9 (636,394) | 1.38 [1.35; 1.41] |
| Missing | 0 (480) | 3.75 [2.39; 5.85] |
| Time period % (*n*) | | |
| 2006–2010 | 24 (1,723,900) | 2.03 [2.01; 2.05] |
| 2010–2013 | 24 (1,688,054) | 1.97 [1.95; 1.99] |
| 2013–2016 | 26 (1,816,273) | 1.89 [1.87; 1.91] |
| 2016–2019 | 26 (1,833,844) | 1.82 [1.80; 1.84] |

[a] Note: IMD deciles are ordered from most deprived (1) to least deprived (10).

CI, confidence interval; IMD, index of multiple deprivation; SD, standard deviation.

being shorter than 2 SDs below the sex-specific mean ($P < 0.001$). Children in the north versus the south of England had a higher probability of short stature, with the highest prevalence in Yorkshire and the Humber (2.18% (2.15 to 2.22)) and the lowest in London (1.57% (1.55 to 1.60)). In individual LAs, short stature ranged from 0.97% (0.85 to 1.10) in Richmond upon Thames in London to 3.92% (3.69 to 4.16) in Blackburn with Darwen in the North West.

There was considerable heterogeneity by ethnicity, with White children being 3 times more likely to have short stature than Black children (1.95% (1.94 to 1.96) versus 0.64% (0.61 to 0.67); $P < 0.001$). Indian children and children in the other ethnic category had the highest prevalence of short stature (2.52% (2.45 to 2.60) and 2.57% (2.51 to 2.64), respectively), which were both significantly higher than White children ($<0.001$). Short stature was also linearly associated with IMD, with prevalence in the most deprived decile being nearly twice that of the least deprived decile (2.56% (2.53 to 2.59) versus 1.38% (1.35 to 1.41); $P < 0.001$). Data for 4 time periods showed that the prevalence of short stature among 4- to 5-year-old children declined between 2006 and 2019 from 2.03% (2.01 to 2.05) to 1.82% (1.80 to 1.84) ($P < 0.001$).

Assessment of change over time by IMD showed that short stature declined across all IMD quintiles between 2006 and 2019. Prevalence declined from 2.67% to 2.21% in the most deprived quintile (quintile 1) and from 1.68% to 1.46% in the least deprived quintile (quintile 5). Full data are presented in S3 Table. A logistic regression model (S4 Table) showed that the decline from 2006 to 2019 within each quintile was significant, although the least deprived quintiles showed slower declines in the prevalence of short stature than the most deprived quintiles, as shown in an Equiplot visualisation (Fig 2).

## Spatial analyses

A total of 326 lower tier LAs existed in England in 2018, 2 of which were aggregated with neighbouring areas due to low numbers (City of London, aggregated with Hackney, and the Isles of Scilly, aggregated with Cornwall), giving a total for analysis of 324 LAs. Short stature clusters adjusted for ethnicity are shown in Fig 3, and cluster descriptions are given in Table 2. All clusters identified in this model were highly significant ($P < 0.001$). There was geographical heterogeneity in short stature prevalence across the country and within regions. A model adjusted for ethnicity found 8 clusters, mostly distributed around the Midlands and North of England, as well as 2 small clusters in London. The clusters with the highest adjusted relative risk (RR) for short stature were in the East Midlands (Leicester; RR: 1.50), East of England (Great Yarmouth; RR: 1.36), and London (Brent; RR: 1.34) and tended to be located in urban areas. Clusters in the North of England had lower RRs but included larger geographic locations and larger total population sizes. There was ethnic heterogeneity between clusters, with very high percentages of White children in some clusters (92% in the North East Lincolnshire cluster) and very low percentages in others (22% in the Newham cluster). Clusters showed relatively high levels of deprivation, with average IMD decile for children in short stature clusters ranging between 2.40 and 4.38 (for comparison, the average IMD for the full sample was 5.08). An unadjusted model as well as a model adjusted for both ethnicity and IMD are presented in S5 Table and S1 and S2 Figs. The general distribution of clusters was similar in the unadjusted and fully adjusted models, although the composition of clusters varied slightly. The Leicester, Great Yarmouth, Tower Hamlets, Gateshead, Rossendale, North East Lincolnshire, and South Staffordshire clusters were represented in each model.

Spatial analyses over the 4 time periods between 2006 and 2019 showed some variation over time in the precise composition of clusters, although the placement of clusters was largely consistent (Fig 4, S6 Table).

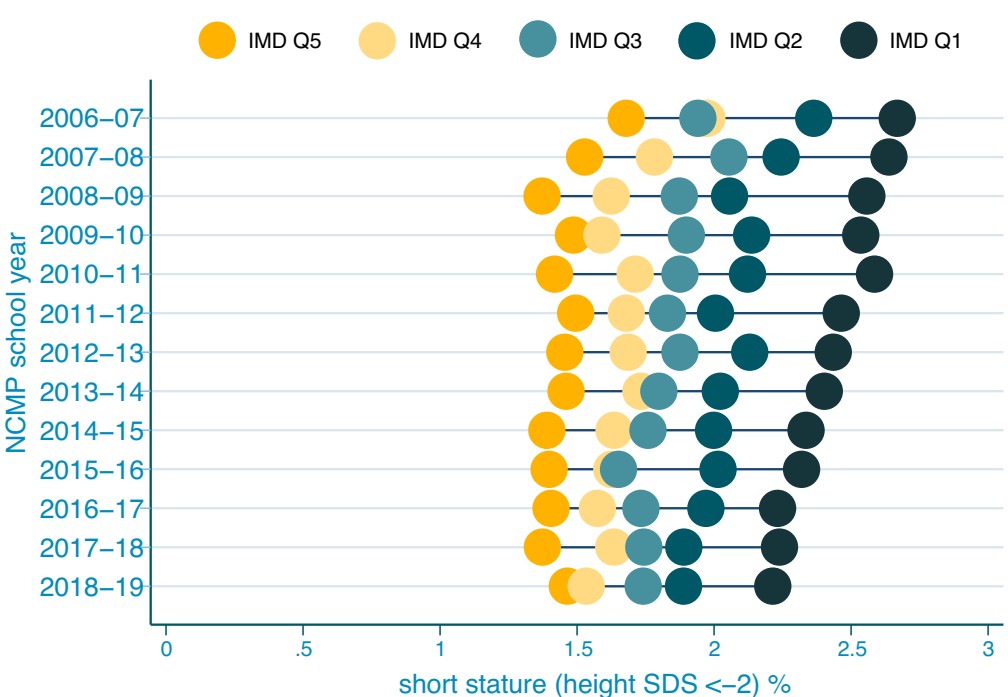

**Fig 2. Equiplot of short stature percentage by IMD quintile and NCMP school year.** IMD Q refers to IMD quintiles, where IMD Q1 is the most deprived quintile, and IMD Q5 is the least deprived quintile. IMD, index of multiple deprivation; NCMP, National Child Measurement Programme; SDS, standard deviation score.

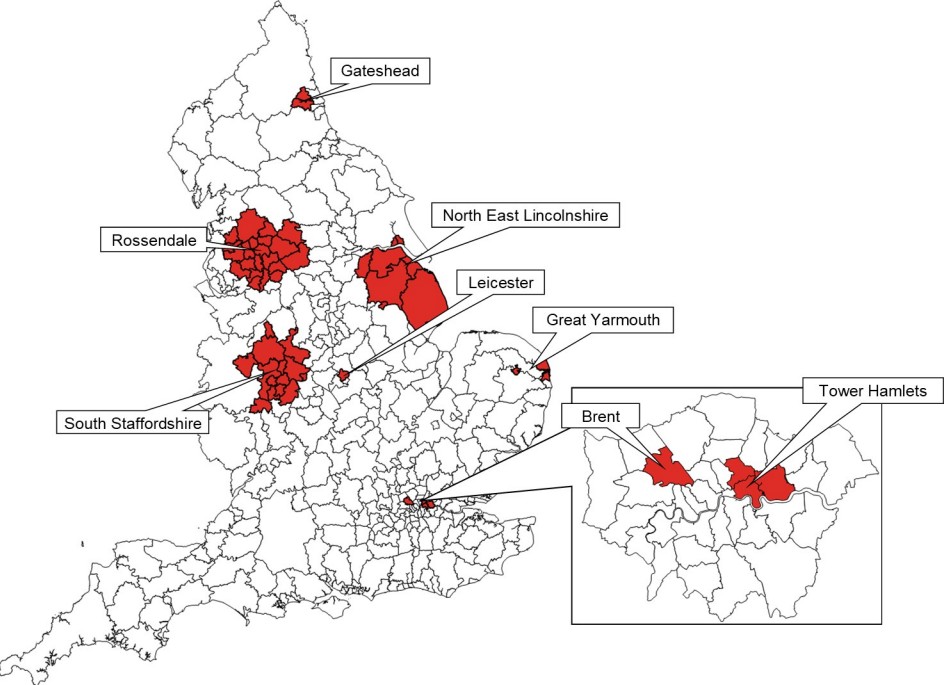

**Fig 3. Short stature clusters in England (London inset) 2006–2019, adjusted for ethnicity (*n* = 5,765,707).** Short stature clusters are in red. Map base layer is shapefile Local Authority Districts (December 2017) Full Clipped Boundaries in Great Britain, published by the Office for National Statistics and available at https://geoportal.statistics. gov.uk/datasets/local-authority-districts-december-2017-full-clipped-boundaries-in-great-britain/explore?location= 55.450000%2C-2.950000%2C5.64&showTable=true.

**Table 2. Short stature clusters in England, 2006–2007 to 2018–2019, adjusted for ethnicity (n = 5,765,707).**

| Cluster[a] | Region | White ethnicity[b] % (n) | Cluster mean (SD) IMD[b] | Population[c] | Short stature % (n) | RR[d] |
|---|---|---|---|---|---|---|
| Leicester | East Midlands | 41 (20,737) | 2.9 (2.0) | 50,088 | 3.1 (1,551) | 1.50 |
| Great Yarmouth, Norwich | East of England | 90 (25,098) | 3.7 (2.5) | 27,793 | 2.8 (783) | 1.36 |
| Brent | London | 25 (35,997) | 3.3 (1.6) | 35,997 | 2.8 (1,002) | 1.34 |
| Tower Hamlets, Newham, Hackney | London | 22 (26,020) | 2.1 (1.1) | 116,778 | 2.6 (3,083) | 1.28 |
| Gateshead, Newcastle upon Tyne | North East | 81 (32,907) | 3.9 (2.9) | 40,757 | 2.6 (1,062) | 1.26 |
| Rossendale, Burnley, Bury, Rochdale, Hyndburn, Blackburn with Darwen, Bolton, Oldham, Calderdale, Pendle, Chorley, Manchester, Salford, Tameside, Wigan, Ribble Valley, Trafford, Bradford, South Ribble, Kirklees, Preston | North West, Yorkshire, and the Humber | 68 (348,059) | 3.7 (2.8) | 515,310 | 2.5 (13,087) | 1.25 |
| North East Lincolnshire, North Lincolnshire, West Lindsey, Kingston upon Hull, East Lindsey, Lincoln | Yorkshire and the Humber, East Midlands | 92 (95,781) | 3.9 (2.7) | 103,569 | 2.5 (2,612) | 1.22 |
| South Staffordshire, Wolverhampton, Cannock Chase, Walsall, Stafford, Sandwell, Telford and Wrekin, Dudley, Lichfield, East Staffordshire, Birmingham, Tamworth, Wyre Forest, Stoke-on-Trent | West Midlands | 62 (272,336) | 3.4 (2.7) | 439,577 | 2.4 (10,618) | 1.18 |

[a] Clusters are referred to in the text by the name of the first LA in the cluster description. These are determined by SaTScan and represent the centre point of the cluster. Clusters are ordered from highest to lowest RR.

[b] Cluster white ethnicity % and mean IMD are derived from NCMP data for children in each cluster.

[c] Cluster population is the total population of NCMP children included in the analysis for each cluster.

[d] No 95% CI is calculated for RR as the method for identifying clusters is data driven, and 95% CIs would be inappropriate.

CI, confidence interval; IMD, index of multiple deprivation; LA, local authority; NCMP, National Child Measurement Programme; RR, relative risk; SD, standard deviation.

## Supplemental analyses

Very short stature ($<-2.67$ SDS) affected 0.36% [0.36 to 0.37] of children, with patterns of prevalence similar to those of short stature ($<-2.0$ SDS) (S7 Table). Spatial analyses of very short stature adjusted for ethnicity identified 6 clusters that broadly matched 6 of the 8 clusters found in the main short stature analysis (S8 Table). A sensitivity analysis of short stature ($<-2.0$ SDS) using the full dataset (n = 7,062,071) (S9 Table) showed high agreement with the main unadjusted model (n = 5,765,707) (S5 Table).

## Discussion

Using a dataset of over 7 million children in England between 2006 and 2019, we found that 1.93% of children aged 4 and 5 had short stature at school entry. Short stature was geographically clustered with higher prevalence in the North and Midlands and lower prevalence in the South. Short stature prevalence ranged from 0.97% in Richmond upon Thames in London to 3.92% in Blackburn and Darwen in the North West, a 4-fold difference that translates into an additional 2,950 children with short stature per 100,000 children starting school. Short stature was highly associated with area-level deprivation, ethnicity, and sex. Deprivation was linearly related to short stature, with the highest prevalence found in the areas with highest deprivation. Yorkshire and the Humber had the highest regional short stature prevalence (2.18%), and London had the lowest (1.57%). However, we identified 2 high prevalence clusters in East and North London, suggesting high heterogeneity within London itself. While there were significant differences in short stature prevalence between children of different ethnicities, there was also high ethnic heterogeneity between clusters, with the proportion of White children ranging

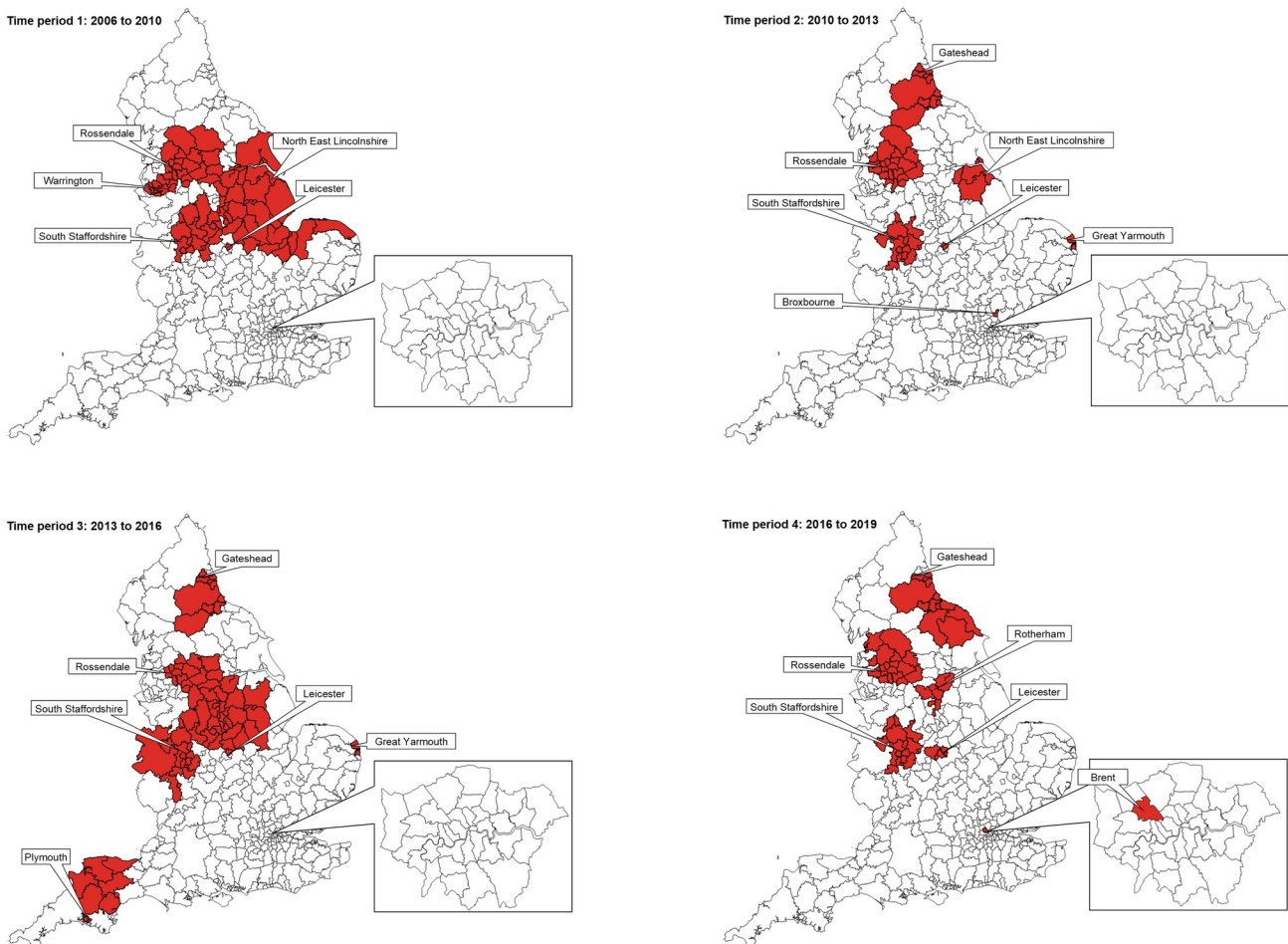

**Fig 4. Spatial analysis over 4 time periods (2006–2019).** Short stature clusters are in red. Map base layer is shapefile Local Authority Districts (December 2017) Full Clipped Boundaries in Great Britain, published by the Office for National Statistics and available at https://geoportal.statistics.gov. uk/datasets/local-authority-districts-december-2017-full-clipped-boundaries-in-great-britain/explore?location=55.450000%2C-2.950000%2C5. 64&showTable=true.

from 22% in the Tower Hamlets cluster to 90% in the Great Yarmouth cluster. Girls were significantly more likely to be short than boys (2.09% versus 1.77%, respectively), which has not previously been reported in high-income settings.

While the national short stature prevalence of 1.93% was broadly as expected, given the normal distribution of heights in a population, the regional differences are striking. Although short stature prevalence declined over time, with larger declines in more deprived IMD deciles, inequalities in the prevalence of short stature by IMD were persistent. This clear association between short stature and deprivation corresponds to a difference in absolute terms of 1,180 per 100,000 reception class children between the most and the least deprived communities (highest and lowest IMD deciles). Collectively, our findings suggest that large numbers of children—particularly those in the most deprived areas of the country—could be failing to reach their full growth potential. Many of these children may in fact be particularly disadvantaged, unhealthy, and failing to thrive.

The link between growth and social conditions is well documented, with growth deemed to be "a mirror of the condition of society" [17]. However, in England, the focus continues to be on overweight and obesity. To our knowledge, this is the first study presenting data on

geographical clustering of short stature in the UK. The World Bank reports an average stunting (short stature) prevalence in high-income countries of 2.8%, but there are very few recent data for international comparison, and the database includes no data from the UK [18]. We found evidence of a clear association between short stature and area-level deprivation. The association holds even for very short stature (height SDS $<-2.67$), whereby prevalence falls monotonically from 0.52 to 0.23 across the IMD deciles (S7 Table). An association between height and deprivation in children of this age has previously been demonstrated [19], but the relationship between deprivation and short stature has not previously been investigated in the UK. There were 2,560 children with short stature per 100,000 in the bottom decile and 1,380 per 100,000 in the top decile (a scale factor of 1.86; Table 1) and clusters identified by spatial analyses all had an IMD below the sample average in unadjusted and ethnicity-adjusted models. Recently published data on child poverty indicators reveal an overlap between areas with high levels of child poverty and the clusters of short stature we identified [20].

Our data demonstrate a striking North–South divide in short stature prevalence in England (Fig 3). A total of 6 of the 8 clusters identified in the ethnicity-adjusted analysis were in the North and Midlands, and there were no clusters outside of London in the South. Spatial–temporal analyses were more complex, but they validated the main results by finding similar patterns of distribution of short stature, where clusters were mainly concentrated in the North and Midlands of the country. This is consistent with national samples of children born between 1920 and 1990 in the UK, in which children born in the North were shorter than children born in the South [21]. A 2015 analysis of around 10,000 White children participating in the Millennium Cohort Study described a "Midlands effect," where children in the Midlands were more likely to be shorter than −1 SDS, compared with children in other regions. The clusters also correspond with the findings of the Marmot Review 10 Years On, which documented regional inequalities in health that have disproportionately affected the North and Midlands. The 2020 Marmot Review also described increased child poverty, greater infant mortality in the most deprived decile, and persisting socioeconomic and regional inequalities in child development and school readiness [22]. While the NCMP data clearly show a relationship between deprivation and short stature, the proximal mechanisms for poor linear growth in high-income settings are not well established. These may include greater infection burden, ambient pollution, poor diet quality, and vitamin D deficiency. Additional mechanisms including adverse childhood experiences, pregnancy outcomes, and epigenetics should be explored in further research.

We found that Indian, Pakistani, and Bangladeshi children had the highest prevalence of short stature, and Black and Mixed children had very low prevalence of short stature. The clusters we identified have high ethnic heterogeneity, with the proportion of White children in high prevalence clusters ranging from over 90% in the North East Lincolnshire cluster to around a quarter in the London clusters. The relationship between ethnicity and linear growth in childhood is complex. The 2006 WHO Multicentre Growth Reference Study indicated that globally, the growth of economically advantaged, breastfed infants, and children of non-smoking mothers is similar [23]. The INTERGROWTH-21st study demonstrated similar findings for fetal and neonatal growth [24]. However, absolute heights and growth patterns between populations are starkly different. The relationship between short stature and ethnicity in the UK is likely to be shaped by complex pathways, including socioeconomic status, immigration patterns, discrimination, and genetic factors.

We found a statistically significantly higher prevalence of short stature in girls compared with boys. Population data on sex differences in short stature are useful, as existing data come from analyses of referral patterns to growth disorder clinics and are therefore subject to selection bias. Evidence from the United States of America suggests that boys are referred more

readily and investigated more thoroughly than girls [25,26]. Short stature is more common in boys than girls in LMICs, and it has been postulated that this is due to higher rates of adverse birth outcomes and increased vulnerability to infection and other morbidity during infancy in boys [27,28]. These data might suggest that this is not a reasonable explanation for sex differences in referral patterns in the UK setting, as the incidence of adverse birth outcomes and childhood infections are higher for boys in the UK [29,30]. It should be noted that the use of UK90 growth references means that rates of short stature in girls and boys will be slightly different to those calculated using WHO standards, which are used in most of the international literature. This may explain some of the discrepancy observed between our results and international findings.

We found a decline in national short stature prevalence between 2006 and 2019. Analyses of British birth cohorts between 1946 and 2001 have previously demonstrated a narrowing of socioeconomic inequalities in child height [31–33]. However, in the context of increasing socioeconomic inequality in England over the last decade [22], the reasons for such a striking decline in prevalence during this period are not clear, especially since the steepest fall was in the most deprived decile (Fig 2). One potential explanation for these findings is that children from the most deprived areas may be benefiting from targeted interventions. This would also explain findings from the 2020 Marmot Review, which found that children from low-income families have demonstrated better development and educational outcomes in low-income areas than children from low-income families in high-income areas [22]. Another hypothesis relates to evidence from diverse populations that childhood obesity is associated with taller stature in childhood (although a shorter adult height). As such, higher levels of overweight and obesity in childhood in deprived children over the period 2006 to 2019 could have driven accelerated linear growth in many children who otherwise might have had short stature. This is supported by published NCMP data, which show static or increased prevalence of overweight/obesity in 5-year-old children in the most deprived IMD quintile, while there is a downward trend in the least deprived quintile [34].

This study had several strengths including the very large national dataset, with coverage of 93% of reception-age children in England over a 13-year period, leading to high precision in our estimates of short stature. Additionally, where previous analyses of associations between height and deprivation in the UK have used data from historical birth cohorts, this analysis of contemporary data allows for more relevant policy inferences for 21st century children. The study also had several limitations. To investigate associations between short stature and deprivation, we used available data on IMD, which is not a measure of individual or household deprivation nor does it completely capture socioeconomic or environmental variables. As such, residual confounding is very likely. The population reference used (UK WHO) is constructed using data from the UK 1990 for the age group analysed, which was, in turn, developed using data from White British children. This may limit its generalisability in children of other ethnicities. Additionally, the NCMP data are cross-sectional, and there is only a single data point for each child, making causal inferences inappropriate.

There are 3 major public health implications of our findings. Firstly, the geographical "hotspots" and the clear socioeconomic gradient demonstrated by these data show that where a child is born, and the environment in which they grow up, both are associated with their height at the age of 4 to 5 years. This should prompt immediate action by local and national government to address the upstream factors that underlie short stature, especially in the areas where we have identified clusters. Moreover, studies of child growth trajectories in LMIC have identified substantial catchup growth by 5 years of age among children who had short stature in infancy [35]. It is therefore possible that the prevalence of short stature is even higher at younger ages in children experiencing poverty in England.

Secondly, the concept of "stunting," or short stature, which broadly reflects socioeconomic determinants of growth faltering, is typically used to describe health inequalities in LMIC but rarely used in high-income settings. In the UK, stunting or short stature is not considered to be a major public health problem because the overall prevalence is much lower than in LMIC. We contest that this difference in approach leads to children with short stature being overlooked in the UK. Our analyses of a large national dataset over the past 13 years, however, identify a clear social gradient and striking regional variation in short stature that should not be ignored.

Finally, while the weights and heights of most 4- to 5-year-old children in England are being systematically measured in the school setting, children with poor linear growth are not being highlighted to their families or general practitioners. This misses a valuable opportunity to identify children whose poor early life growth may be associated with poor health and delayed neurodevelopment in childhood and chronic disease and all-cause mortality in adulthood [36]. UK guidance recommends referral for children with height below the 0.4th centile ($<-2.67$ SDS). This guideline is stricter than other European countries and has low sensitivity for detecting growth disorders (around 30%) [37]. The combination of height SDS, parental heights, and decreased growth rate can be used effectively for growth monitoring with optimal cutoff levels. However, this is currently not possible in the UK as parental heights are not routinely assessed, and repeat measurements are not undertaken. Around 60% to 80% of short children ($<-2.0$ SDS) are estimated to have no identifiable aetiology following review [38]. We argue that screening should start earlier in life than the current NCMP programme, since school-age measurements may be too late to mitigate many of the factors underlying short stature. It is now well established that the "first 1,000 days" (conception to age 2 years) is the period during which linear growth and neurodevelopment are most sensitive to environmental modification. We therefore propose that height should be systematically screened at younger ages in the UK, in line with other European countries. For example, children in the Netherlands and Finland have heights and weights routinely measured at least 10 times in the first 1,000 days [39]. The National Screening Committee and the Health and Social Care Committee have both recommended growth screening in early childhood, yet no such programme currently exists [40, 41]. Earlier, systematic, nationwide screening and identification of linear growth faltering could trigger timely referral for investigations to identify those with underlying medical disorders and an opportunity for psychosocial and educational intervention prior to school entry in those without underlying medical problems.

## Supporting information

**S1 Text. STROBE Statement.** Checklist of items that should be included in reports of observational studies. STROBE, Strengthening the Reporting of Observational Studies in Epidemiology.
(DOCX)

**S2 Text. Proposal.** Barts Charity grant proposal research analysis plan (2018).
(DOCX)

**S3 Text. SaTScan model specification.** Table A: Height (cm) by ethnicity regression results, boys ($n$ = 2,946,560). Table B: Height (cm) by ethnicity regression results, girls ($n$ = 2,819,147). Table C: Height (cm) by ethnicity and IMD regression results, boys ($n$ = 2,946,560). Table D: Height (cm) by ethnicity and IMD regression results, girls ($n$ = 2,819,147). IMD, index of multiple deprivation.
(DOCX)

**S1 Table. Ethnicity and IMD missingness by sample characteristics.** IMD, index of multiple deprivation.
(DOCX)

**S2 Table. Population sex, Government Office Region, and ethnicity in children, 2011–2012 (Census 2011 and ONS population projections).**
(DOCX)

**S3 Table. Percentage of children with short stature (<−2.00 SDS) by NCMP school year and IMD decile ($n$ = 7,061,591).** IMD, index of multiple deprivation; NCMP, National Child Measurement Programme; SDS, standard deviation score.
(DOCX)

**S4 Table. Logistic regression of short stature (<−2.00 SDS) by IMD and year, including IMD#year interaction ($n$ = 7,061,591).** IMD, index of multiple deprivation; SDS, standard deviation score.
(DOCX)

**S5 Table. Short stature (<−2.00 SDS) clusters, unadjusted and adjusted for both ethnicity and IMD ($n$ = 5,765,707).** IMD, index of multiple deprivation; SDS, standard deviation score.
(DOCX)

**S6 Table. Time period analysis of short stature (<−2.00 SDS) ($n$ = 7,062,071). SDS, standard deviation score.**
(DOCX)

**S7 Table. Very short stature (<−2.67 SDS) prevalence by sample characteristics ($n$ = 7,062,071). SDS, standard deviation score.**
(DOCX)

**S8 Table. Very short stature (<−2.67 SDS) clusters in England, 2006–2007 to 2018–2019, unadjusted, adjusted for ethnicity, and adjusted for ethnicity and IMD ($n$ = 5,765,707).** IMD, index of multiple deprivation; SDS, standard deviation score.
(DOCX)

**S9 Table. Short stature (<−2.00 SDS) clusters, unadjusted, full sample ($n$ = 7,062,071). SDS, standard deviation score.**
(DOCX)

**S1 Fig. Short stature clusters in England (London inset) 2006–2007 to 2018–2019, unadjusted.**
(DOCX)

**S2 Fig. Short stature clusters in England (London inset) 2006–2007 to 2018–2019, adjusted for ethnicity and IMD. IMD, index of multiple deprivation.**
(DOCX)

## Author Contributions

**Conceptualization:** Joseph Freer, Joan K. Morris, Robert Walton, Leo Dunkel, Helen L. Storr, Andrew J. Prendergast.

**Data curation:** Caroline Hancock, Andrew J. Prendergast.

**Formal analysis:** Joanna Orr, Joan K. Morris, Caroline Hancock.

**Funding acquisition:** Joseph Freer, Joan K. Morris, Robert Walton, Leo Dunkel, Helen L. Storr, Andrew J. Prendergast.

**Investigation:** Joanna Orr, Joseph Freer, Joan K. Morris, Robert Walton, Leo Dunkel, Helen L. Storr, Andrew J. Prendergast.

**Methodology:** Joanna Orr, Joan K. Morris, Andrew J. Prendergast.

**Project administration:** Andrew J. Prendergast.

**Software:** Caroline Hancock.

**Supervision:** Joan K. Morris, Andrew J. Prendergast.

**Writing – original draft:** Joanna Orr, Joseph Freer, Andrew J. Prendergast.

**Writing – review & editing:** Joanna Orr, Joseph Freer, Joan K. Morris, Caroline Hancock, Robert Walton, Leo Dunkel, Helen L. Storr, Andrew J. Prendergast.

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
