## [Editor Report · Decision Letter 0]

6 Apr 2021

Dear Dr Orr, 

Thank you for submitting your manuscript entitled "A spatial analysis of child stunting in England between 2006-2019" for consideration by PLOS Medicine.

Your manuscript has now been evaluated by the PLOS Medicine editorial staff and I am writing to let you know that we would like to send your submission out for external peer review.

Please re-submit your manuscript within two working days, i.e. by April 8, 2021.

Kind regards,

Beryne Odeny

Associate Editor

PLOS Medicine

---

## [Decision Letter · Decision Letter 1]

10 Jun 2021

Dear Dr. Orr,

Thank you very much for submitting your manuscript "A spatial analysis of child stunting in England between 2006-2019" (PMEDICINE-D-21-01543R1) for consideration at PLOS Medicine. 

[LINK]

In light of these reviews, I am afraid that we will not be able to accept the manuscript for publication in the journal in its current form, but we would like to consider a revised version that addresses the reviewers' and editors' comments. Obviously we cannot make any decision about publication until we have seen the revised manuscript and your response, and we plan to seek re-review by one or more of the reviewers. 

We expect to receive your revised manuscript by Jul 01 2021 11:59PM. Please email us (plosmedicine@plos.org) if you have any questions or concerns.

We look forward to receiving your revised manuscript. 

Sincerely,

Beryne Odeny, 

PLOS Medicine

plosmedicine.org

1) Please revise your title according to PLOS Medicine's style. Please place the study design (e.g., "A retrospective cohort study”) in the subtitle (ie, after a colon). 

2) In the abstract Methods and Findings:

a) Please structure your abstract using the PLOS Medicine headings (Background, Methods and Findings, Conclusions).

b) Please combine the Methods and results sections into one section, “Methods and findings”. 

c) Please ensure that all numbers presented in the abstract are present and identical to numbers presented in the main manuscript text.

d) Please quantify the main results (with 95% CIs and p values).

e) Please include the important dependent variables that are adjusted for in the analyses.

f) In the last sentence of the Abstract Methods and Findings section, please describe the main limitation(s) of the study's methodology.

g) Please address the study implications without overreaching what can be concluded from the data; the phrase "In this study, we observed ..." may be useful. Mention only specific implications substantiated by the results.

3) Author summary - At this stage, we ask that you include a short, non-technical Author Summary of your research to make findings accessible to a wide audience that includes both scientists and non-scientists. The Author Summary should immediately follow the Abstract in your revised manuscript. This text is subject to editorial change and should be distinct from the scientific abstract. Please see our author guidelines for more information: https://journals.plos.org/plosmedicine/s/revising-your-manuscript#loc-author-summary.

4) Did your study have a prospective protocol or analysis plan? Please state this early in the Methods section. 

5) Your study is observational and therefore causality cannot be inferred. Please remove language that implies causality, such as impact. Refer to associations instead. For example, in the discussion section, replace the statement, “… both have a clear impact on height at the age of 4-5” with “… are associated with height at the age…”

6) Please add the following statement, or similar, to the Methods: "This study is reported as per the Strengthening the Reporting of Observational Studies in Epidemiology (STROBE) guideline (S1 Checklist)." The STROBE guideline can be found here: http://www.equator-network.org/reporting-guidelines/strobe/ . When completing the checklist, please use section and paragraph numbers, rather than page numbers.

7) In the Methods and Results section:

a) Please provide 95% CIs and p values for all estimates.

b) When a p value is given, please specify the statistical test used to determine it.

c) Please present numerators and denominators for percentages, at least in the Tables

8) For your Tables and figures, please define the abbreviations such as SD, IMD, NCMP, SE, CI

9) Please use the "Vancouver" style for reference formatting and see our website for other reference guidelines. For example, six names should appear before et al. Please ensure that journal name abbreviations match those found in the National Center for Biotechnology Information (NCBI) databases, and are appropriately formatted and capitalized. https://journals.plos.org/plosmedicine/s/submission-guidelines#loc-references. Please ensure that weblinks are current and accessible to date.

a) Six names should appear before et al, in the reference section. Please correct refs # 20, 29, 31

b) Please include access dates for all weblinks and ensure that all weblinks are current and accessible.

10) Please include line numbers in you next revision

Comments from the reviewers:

Reviewer #1: 

This is a very important study and the first to my knowledge that height of children from the NCMP program has been analysed separately from obesity alone and that has powerful socio-economic implications. 

The study is well conducted appropriately statistically analysed and very well presented including the extra supplemental material, although the statistical process is one of spatial analysis, 'regional analysis' might be a more accessible term for to use for the title.

I am uncomfortable with the use of the term 'stunting' here. Although the WHO does use this for height of children below -2 standard deviations it also includes in the definition of not achieving their full adult high potential. That is very difficult to infer from this study given the information presented here, hence in my view the term 'short stature' is a more preferable term to stunting as there is no evidence of a long term growth decrement and a reduction of adult height compared with genetic or familial expectations. The simple explanation could be different growth patterns. We know already that Afro-Caribbean childen are more physiologically advanced than Caucasians.

I have no additional comments to make about the ethical or analytical processes. These appear to be well done and I am not an expert in spatial analysis.

Data of the proportions of the sample each of the region are presented in table 1 together with the ethnicity breakdown. It would be helpful for comparative purposes if the figures from the 2011 population survey were included to validate these values.

The changes in the IMD quintile over time as presented in figure 2 is very important information and perhaps reasons that this has occurred could be explored in more detail in the discussion section other than putting this down to obesity. Presumably with the NCMP data you can affirm or deny that conclusion.

Height determinations are made according to the UK 1990 British references. Reference 11 is correct but WHO data are used until age 4 then UK 1990 thereafter. These are built exclusively on the white British population therefore further adjustment needs to be made to represent all the ethnic minorities present. It is little surprising that those of Asian background are significantly shorter and those of Afro-Caribbean taller. That discussion needs to be expanded. The presentation of the proportion below the second centile by ethnic background is important. However, there are no references to known heights of adults from relative ethnic backgrounds presented and perhaps a further sub-analysis based on this information might be helpful for further interpreting the findings. Reference to the country of origin might need to be made.

Furthermore, the data used to produce the 1990 charts was collected from a number of sources and at that stage there already was a north south divide evident in heights of children within the UK. The 1995 Freeman and Cole paper in Archives of Disease in Childhood 1995; 73: 17-24 discusses differences in heights across the country with Newcastle infants being lighter and Leeds school children being shorter than those in the south. The male-female disparity was different however.

On page 17 there is a comment that those without an identifiable cause of stunting would be labelled as idiopathic short stature. However, a simple height measurement below the second centile cannot really define either idiopathic short stature or stunting. The value of height screening has been much debated but there is significant evidence in particular for Dutch and Finnish studies as discussed in the paper that a stepwise approach can be taken to identify short children with significant pathology but often requires repeat measurements and knowledge of the parents' heights. Hence, if an argument was constructed to use height measurement to detect socioeconomic deprivation and to detect pathology more often than the NCMP programme could allow, then the discussion needs to reflect that. I don't think that the final line of the abstract conclusion, 'Many children in the most deprived areas of the country may be failing to reach their full growth potential' can be fully justified based on the information presented here without additional expansion. Overall, though this is an important paper and needs to be presented in the public domain.

Reviewer #2: The authors describe a spatial analysis of stunting in English children aged 5, based on the NCMP. The paper is excellent, with an extremely large dataset imaginatively analysed and carefully interpreted. I have some mainly minor comments on the study analysis, presentation and interpretation.

1. I should declare my conflict of interest, having constructed (with Mike Preece and Jenny Freeman) the British 1990 height reference, which at age 5 equates to the UK-WHO reference. The stunting rate should be close to 2.28%, the proportion below -2 SDS, so the paper provides a large-scale validation of our chart construction skills.

2. I am pleased to see that the mean stunting rate of 1.93% is close to the expected value, though the apparent sex difference in rates probably reflects a bias in the reference - see later.

3. Abstract. Note that the UK-WHO is a growth reference, not a growth standard.

4. When describing the national state school coverage, it would be useful to say what percentage of children aged 5 are in state schools.

5. What is a "suppressed" version of the NCMP dataset?

6. Why was the cut-off of ±5 SDS used for data cleaning, as opposed to say ±4?

7. The definition of severe stunting used in the paper is wrong (page 5). The cut-off is -2.67 not -2.65 SDS, following the convention proposed by me (Cole TJ. Do growth chart centiles need a face lift? BMJ 1994;308:641-642). Thus the nominal severe stunting rate is 0.38% rather than 0.40%. The whole sentence needs redrafting, as it is not true to say it is "often" used as a referral cut-off, it is _the_ cut-off for defining short stature in the UK. See for example https://www.rcpch.ac.uk/sites/default/files/Boys_2-18_years_growth_chart.pdf. This relates to comments later about idiopathic short stature - they should relate to the -2.67 not the -2 cut-off. 

8. The Statistical Analyses section does not include a sentence specifying the level of significance being used. Despite this p-values appear throughout the text, mainly p < 0.001. I suggest omitting most if not all of the p-values, as the sample size is enormous and the standard errors tiny, ensuring significance even for very small differences. The confidence intervals provide the same information more effectively.

9. In Figure 2 it would be useful to reverse the IMD legend so it's in the same direction as the graphic. What does the title "Equiplot (13)" mean?

10. The cluster results in Figure 3 are very interesting, particularly the position of Leicester, which tops the list but appears as a tiny blob in the figure. I suggest labelling the clusters, primarily for non-British readers. It would also be worth discussing in a bit more detail exactly why Leicester should achieve this dubious distinction. Clearly it has a high proportion of Asian children, but ethnicity is adjusted for so it ought not to be the explanation. Has Leicester featured prominently in other indicators of deprivation?

11. The title for Table 2 looks odd, saying it's adjusted both for ethnicity and for ethnicity and IMD. I suspect just the former is true. The table gives the stunting rate and RR to two decimal places, which is appropriate, but giving the white ethnicity percentage to two decimal places is not. It aims to distinguish between clusters like Brent, where 25% of children are white, and Great Yarmouth, where 90% are white. These percentages can given as whole numbers, and greater precision is unnecessary. This should apply throughout the text as well as the table. See my guidelines on the presentation of numerical information at http://adc.bmj.com/cgi/content/full/archdischild-2014-307149. Also in Table 2 I suggest omitting the log-likelihood ratio and p-value columns, as they are respectively uninterpretable and uninformative. Note too that log-likelihood ratios to two decimal places are grossly over-precise. Table 1 would also be better with fewer decimal places.

12. Coming back to the sex difference, the stunting rates by sex are 2.09% for girls and 1.77% for boys. The nominal rate is 2.28%, so it's not so much that girls are high as that boys are low. But in fact the differences are small - the corresponding z-scores are -2.10 and -2.04, which equate to 2-5 mm deviation from the expected cut-offs. The Discussion states that this sex difference has not previously been reported in a high income setting, but it is almost certainly not generalisable. It is much more likely a bias in the reference as originally constructed. It may be worth mentioning the BMJ letter that we published soon after publishing the reference: Preece MA, Freeman JV, Cole TJ. Sex differences in weight in infancy - Published centile charts for weight have been updated. BMJ 1996;313:1486. It referred to infancy not age 5, but it indicates the care needed to avoid building a sex bias into the reference.

13. The second paragraph of the Discussion refers to idiopathic short stature, but as mentioned above it needs to be made clear that this diagnosis formally requires evidence of severe stunting, i.e. height SDS < -2.67. The same idea is picked up at the foot of page 17, where it is suggested that stunting is not recognised as an indicator of deprivation. 

14. The scale of the dataset leads to some very powerful trends. The first paragraph of the Discussion states—correctly—that "Deprivation was linearly related to stunting", yet this statement holds even when applied to severe stunting, where in supplemental table C the severe stunting rate by IMD falls, perfectly monotonically, from 0.55 to 0.24 across the ten categories. I find this astonishing, and it calls to mind the famous phrase of James Tanner: "Growth as a mirror of the condition of society" (published as Tanner JM. Growth as a mirror of the condition of society: secular trends and class distinctions. Acta Paediatr Jpn 1987;29:96-103). This paper would be worth citing.

15. The reference to the dual burden and the suggestion that increasing obesity is fuelling a reduction in stunting in the more deprived is well made.

16. The paper lists its limitations, but I disagree that the last two mentioned are valid limitations. "As such, we will miss children with faltering growth whose height is crossing centiles and children whose height is above -2 SDS but who are below their target height as predicted by parental heights." Cross-sectional data cannot be expected to provide evidence of centile crossing, and the chosen definition of stunting does not involve parental height. I recommend removing the sentence.

17. A few minor comments on the Supplemental materials. The regression results should specify the units as cm. I suggest omitting the SE columns as they are so tiny (and the SE for the constant in Table A3 needs some attention). I found the table titles odd, a mix of letters and numbers with no obvious structure.

Tim Cole

Reviewer #3: This is a well-conducted spatial analysis of child stunting in England between 2006-2019. The study design, dataset, statistical methods and analyses, and presentation (tables and figures) and interpretation of the results are mostly adequate and of a good standard. The stats analyses in the supplementary information are very thorough and comprehensive with missing data issue particularly well addressed. Only a couple of minor points needing attention.

1) Table 2. Not clear what exactly was adjusted for. In the title, it says 'adjusted for ethnicity and adjusted for ethnicity and IMD' but in the table, there is only a line saying 'Adjusted for ethnicity'. Can authors please clarify this?

2) Time trends of stunting hot spots are striking as shown in Supplementary Figure F. Maybe authors can move Figure F into the main paper as it's of public health interest and give it a bit more discussion on the changes over time and the reasons behind these changes.

[LINK]

---

## [Decision Letter · Decision Letter 2]

26 Jul 2021

Dear Dr. Orr,

Thank you very much for re-submitting your manuscript "Regional differences in short stature in England between 2006-2019: A cross-sectional analysis from the National Child Measurement Programme" (PMEDICINE-D-21-01543R2) for review by PLOS Medicine.

I have discussed the paper with my colleagues and the academic editor and it was also seen again by three reviewers. I am pleased to say that provided the remaining editorial and production issues are dealt with we are planning to accept the paper for publication in the journal.

[LINK]

We look forward to receiving the revised manuscript by Aug 02 2021 11:59PM.   

Sincerely,

Beryne Odeny, 

Associate Editor 

PLOS Medicine

plosmedicine.org

Requests from Editors:

1) Thank you for providing your STROBE checklist. Please replace the page numbers with paragraph numbers per section (e.g. "Methods, paragraph 1"), since the page numbers of the final published paper may be different from the page numbers in the current manuscript.

Comments from Reviewers:

Reviewer #1: This is now a fantastic paper. Congratulations to the authors and for all the revisions. I recommend publication.

Only 2 minor changes suggested:

Abstract P2 line 11 : should be ....heights from a total of 7xxxxxx children were analysed.

Discussion p16 line 3:A 2015 analysis ...... children in the midlands were most likely to be to be shorter than SDS-1. I'm not completely clear what this means

Good luck!

Reviewer #2: The authors have responded very positively to my suggestions. I have a few more minor comments for them to consider.

1. The Discussion refers to the stunting rate in an international context, but it should be remembered that the British 1990 reference is not widely used internationally. This needs to be emphasised, and it would also be worth giving the rate as based on the WHO standard/reference. For girls the -2 SDS cut-off at 60 months is exactly the same (99.91 cm) for UK90 and WHO, but for boys it is 100.7 cm for WHO as against 100.5 for UK90. With a higher boys WHO cut-off there ought to be more WHO stunting, and I estimate the boys rate with WHO as 1.93% as against 1.77% with UK90. You can probably calculate it directly. 

2. This comparison between UK90 and WHO is also useful for interpreting the sex difference, in that the UK90 and WHO cut-offs are broadly the same so that perhaps the girls' excess is more genuine that I previously thought. Note that reference 25 (mentioned on page 16) refers explicitly to weight in infancy, and hence it is not directly relevant to height at age 5 - I referred to it just to highlight how tricky it can be to estimate growth centiles.

3. I note that the acronym SDS is used with the -2.67 cut-off but not with -2.0. I suggest harmonising its usage.

4. In Table 2 just one decimal place would be better for the cluster means.

5. The phrase 'short-statured children' appears on page 3, which might be better as 'children with short stature'. The later bullet point starting 'Short stature' should either omit 'also' or else reverse the sentence.

6. The maps of the clusters look very good, but I wondered if they could use colour more effectively, e.g. by adding a colour scale to reflect the % of white children in each cluster - just a thought, but it would emphasise how heterogeneous the ethnicity mix across the clusters. 

Tim Cole

Reviewer #3: Many thanks authors for their effort to improve the manuscript. I am satisfied with the response and revision. No further issues needing attention.

[LINK]

---

## [Editor Report · Decision Letter 3]

5 Aug 2021

Dear Dr Orr, 

On behalf of my colleagues and the Academic Editor, Dr. Zulfiqar A. Bhutta, I am pleased to inform you that we have agreed to publish your manuscript "Regional differences in short stature in England between 2006-2019: A cross-sectional analysis from the National Child Measurement Programme" (PMEDICINE-D-21-01543R3) in PLOS Medicine.

PRESS

Sincerely, 

Beryne Odeny 

Associate Editor 

PLOS Medicine